# Inertial Measurement Unit Sensor-to-Segment Calibration Comparison for Sport-Specific Motion Analysis

**DOI:** 10.3390/s23187987

**Published:** 2023-09-20

**Authors:** Mitchell Ekdahl, Alex Loewen, Ashley Erdman, Sarp Sahin, Sophia Ulman

**Affiliations:** 1Scottish Rite for Children, Frisco, TX 75034, USA; alex.loewen@tsrh.org (A.L.); ashley.erdman@tsrh.org (A.E.); sarp.shaheen@gmail.com (S.S.); sophia.ulman@tsrh.org (S.U.); 2Department of Orthopaedic Surgery, University of Texas Southwestern Medical Center, Dallas, TX 75390, USA

**Keywords:** inertial sensors, motion analysis, functional calibration, wearable sensors, sports biomechanics

## Abstract

Wearable inertial measurement units (IMUs) can be utilized as an alternative to optical motion capture as a method of measuring joint angles. These sensors require functional calibration prior to data collection, known as sensor-to-segment calibration. This study aims to evaluate previously described sensor-to-segment calibration methods to measure joint angle range of motion (ROM) during highly dynamic sports-related movements. Seven calibration methods were selected to compare lower extremity ROM measured using IMUs to an optical motion capture system. The accuracy of ROM measurements for each calibration method varied across joints and sport-specific tasks, with absolute mean differences between IMU measurement and motion capture measurement ranging from <0.1° to 24.1°. Fewer significant differences were observed at the pelvis than at the hip, knee, or ankle across all tasks. For each task, one or more calibration movements demonstrated non-significant differences in ROM for at least nine out of the twelve ROM variables. These results suggest that IMUs may be a viable alternative to optical motion capture for sport-specific lower-extremity ROM measurement, although the sensor-to-segment calibration methods used should be selected based on the specific tasks and variables of interest for a given application.

## 1. Introduction

Lower limb kinematic assessments are an integral part of sports biomechanics research, specifically for investigations focused on injury prevention, performance analysis, and/or return-to-play evaluations. Optical motion capture systems are widely considered the gold standard for these assessments [1]; however, the limitations of these systems restrict their use in sports research. Most notably, they require a large laboratory space and cannot be transported for testing in the field. In addition, these systems are relatively expensive and are susceptible to marker occlusion during highly dynamic sports-related tasks [2]. Recently developed wireless wearable sensors known as inertial measurement units (IMUs) are a portable, relatively low-cost alternative to optical motion capture for studying lower limb kinematics outside of the lab setting [3,4,5]. These sensors have been applied to sports kinematics analysis in many ways [6], including injury risk assessment in healthy subjects [7] and functional evaluation in a clinical setting following an ACL injury [8]. IMUs contain a three-dimensional (3D) accelerometer, gyroscope, and magnetometer, and the motion of the IMUs can be tracked over time through the integration of the acceleration and angular velocity signals. Other methods of sensor orientation estimation also exist, such as extended Kalman filtering, in which multiple signals from the IMUs are combined to improve estimation accuracy [9].

IMUs have an internal coordinate system that is independent of the anatomical coordinate system of the body segment on which it is placed (Figure 1) [10]. The two coordinate systems can be aligned through a process known as sensor-to-segment calibration [11]. During this process, the signals from an IMU are translated to the anatomical reference frame to track the motion of the segment rather than the sensor itself. Unlike motion capture, a widely accepted method does not exist for defining the anatomical reference frame for a given segment. Other groups have developed sensor-to-segment calibration methods for use in gait analysis and upper-extremity kinematics; however, these methods are not known to have been validated for joint angle estimation in dynamic sports-specific tasks [12,13]. Sports-related movements present additional challenges for kinematic assessment relative to normal gait, including increased joint range of motion and multi-planar motion, as well as increased signal noise due to rapid foot contact and skin artifact. These additional challenges represent potential sources of error in joint angle estimation that are either not present or are reduced in tasks such as gait. Therefore, this study was intended to identify methods of IMU sensor-to-segment calibration that may be used in sports-related lower limb kinematic assessments.

One previously described method of calibration requires positioning each IMU such that the axes of the sensors are aligned with the axes of the respective segments [14]. For example, a sensor would be placed on the shank such that one axis of the sensor is aligned with the tibia. This approach is highly user-dependent as it requires skilled, precise placement of the sensors on each segment and consistency across participants. This also may not be suitable for some applications when sports-specific equipment like shin guards or shoulder pads are covering the desired IMU placement area. A second method uses a series of relatively arbitrary calibration movements and applies biomechanical assumptions of the joints to estimate anatomical axes [15,16]. Most commonly, this method is applied to joints such as the knee, which can be approximated as a hinge joint that only moves in the sagittal plane. This method is not suitable for other joints with significant motion in multiple planes and does not allow for the assessment of common sports injury risk factors such as dynamic knee valgus. The third and most common method of calibration involves the collection of a series of static postures or predefined movements with motion limited to a single plane and does not require any biomechanical assumptions [13,17,18]. The accelerometer signal during static postures is used to identify the orientation of each IMU relative to the gravity vector, and the gyroscope signal during functional tasks is used to estimate the anatomical plane of motion. By combining data from multiple static postures or dynamic tasks, the anatomical axes of each body segment can be estimated with respect to the IMUs.

Commercially available wearable IMU systems have been demonstrated to measure joint angles accurately during sports tasks and require simple calibration procedures specified by the manufacturer [19,20,21]. However, more advanced systems are expensive, and in some instances, the calibration and computation procedures are provided separately at an additional cost [22]. Thus, the overarching goal of this study was to provide details of how to independently calibrate an IMU system for the collection of dynamic movements. Specifically, the purpose of this study was to assess the accuracy of static and functional sensor-to-segment calibration methods for use in sports biomechanics research and to compare different functional calibration movements across a series of tasks commonly used in sports-specific lower limb kinematic assessments. To our knowledge, IMU sensor-to-segment calibration has not been investigated for use in kinematic assessments involving highly dynamic sports-related tasks. We believe the decreased signal-to-noise ratio and increased joint range of motion that is inherent in sports kinematic assessments warrant validation of previously described sensor-to-segment calibration methods for use in sports research. This is an exploratory study and is, to our knowledge, the first to assess the validity of these sensor-to-segment calibration methods in the context of sports kinematics assessments.

## 2. Materials and Methods

### 2.1. Participants

A convenience sample of eleven healthy volunteers participated in this study (four females, 24.5 ± 3.5 years) and completed a single visit to a motion analysis laboratory. To be eligible, participants were required to self-report no orthopedic condition or prior injury (within the past six months) that would limit their ability to perform the required tasks. This study was approved by the University of Texas Southwestern Institutional Review Board (Protocol Number: 082010-134, Approval Date: 2 February 1993), and all participants provided informed written consent prior to initiating testing procedures. For testing, participants were asked to wear comfortable attire and their personal athletic footwear.

### 2.2. Setup and Equipment

Seven wearable wireless Delsys Trigno Avanti sensors (Delsys Inc., Natick, MA, USA) were placed on body segments of interest as shown in Figure 2. One sensor was placed on each segment: The sacrum, anterior thighs, anterior shanks, and the dorsal side of the feet. Each sensor was tightly secured with self-adherent elastic wrap and athletic tape to minimize noise in the data due to skin artifacts. Each sensor contained an IMU that consists of a tri-axial accelerometer (±16 g), gyroscope (±2000 dps), and magnetometer. All acceleration and angular velocity data were collected at 240 Hz. Data from the magnetometer could not be accessed via third-party software and were not used for this study.

Optical motion capture was used as a gold standard comparison to the IMUs. Retroreflective markers were placed on bony landmarks on the trunk, pelvis, and lower extremities according to a modified Cleveland Clinic marker set [23]. Marker placement was similar to Ulman et al. [24] with the addition of bilateral patella markers, as well as posterior superior iliac spine markers instead of a sacral marker to allow for IMU placement. A 14-camera motion capture system (Vicon Motion Systems Ltd., Denver, CO, USA) was used to collect marker data sampling at 240 Hz. Each camera emits infrared light, which is reflected by the markers, and subsequently detected by the camera to estimate the position of the markers relative to the camera [25]. By collecting marker position data from multiple cameras, the motion capture system is able to reconstruct the position of each marker in 3-dimensional space. All motion capture and IMU data were collected using Vicon Nexus 2.11 software and time synchronized with a Delsys Trigger Module (Delsys Inc., Natick, MA, USA).

### 2.3. Testing Procedure

Each participant performed two static calibration poses and six functional calibration movements. The standing and seated static positions were used to determine the orientation of each sensor relative to gravity utilizing the accelerometer data from the IMUs. The dynamic calibration movements were adapted from Lebleu et al. [13] and were intended to be easily reproducible. The reader is referred to Lebleu et al. [13] for an illustration of each dynamic calibration method. Each movement involves significant motion in the sagittal plane while minimizing motion in the coronal and transverse planes. Each functional calibration movement was demonstrated by a staff member and was performed twice by the subject. The verbal instructions for each calibration trial are listed in Table 1.

Following the calibration trials, each participant completed a series of test movements. Participants completed a straight-line gait followed by seven functional tasks: Overhead Squat, Heel Touch, Single Leg Hop, Drop Vertical Jump, Lateral Shuffle, Deceleration, and 45° Cut. Descriptions and verbal instructions for each functional task are described in Ulman et al. [26]. Participants were instructed to begin and end each trial in a neutral standing position to ensure the sensors were stationary. Two repetitions of each task were performed, and additional repetitions were collected if a task was performed incorrectly at the discretion of the research staff. One representative trial for each task was selected for analysis based on a performance metric: Lowest squat depth for the Overhead Squat and Heel Touch, furthest jump for the Single Leg Hop, highest jump for the Drop Vertical Jump, and fastest entry velocity for the Lateral Shuffle, Deceleration, and 45° Cut.

### 2.4. Data Processing

Motion capture data were processed in Vicon Nexus 2.11 software. A Woltring filter with predicted mean square error of 10 mm^2^ was applied to all marker trajectories, and force plate data were filtered with a fourth-order Butterworth filter with a lowpass cutoff frequency of 10 Hz (Overhead Squat, Heel Touch) or 16 Hz (Single Leg Hop, Drop Vertical Jump, Lateral Shuffle, Deceleration, 45° Cut). Lower extremity joint angles as well as segment orientation relative to the laboratory coordinate system were computed using a custom 6-degree-of-freedom model in MATLAB 2022a (MathWorks, Natick, MA, USA) based on 3D marker position data.

A custom MATLAB script was used to process all IMU data from the calibration movements and functional tasks (MATLAB 2022a, Natick, MA, USA). During the standing static trial, it was assumed that the transverse axis of each segment was parallel to the gravity vector measured by the IMUs. The seated static position was intended to rotate all segments in the sagittal plane relative to the standing static position, and the gravity vector measured by each IMU was assumed to be in the sagittal plane of the respective segment. The sagittal axis of each segment was estimated by taking the cross-product of the gravity vector measured during the seated static and the gravity vector measured during the standing static.

Principal component analysis (PCA) was performed on the IMU data from each of the functional calibration movements similar to Lebleu et al. [13]. PCA is widely used to reduce the dimensionality of a dataset by identifying a new basis of lower dimension that still describes most of the variation in the data. When applied to 3-dimensional inertial data, the first principal component defines the direction in which most of the motion occurs, and the first two principal components define the plane in which most of the motion occurs [27]. For the three gait-based calibration movements (Slow Gait, Gait, and Fast Gait), PCA was run on the accelerometer data. It was assumed that most of the variance in the accelerometer signal would occur in the sagittal plane of each segment, therefore, the first and second principal component vectors defined the estimated sagittal plane. The third principal component, which is orthogonal to the first two, was assumed to be parallel to the sagittal axis of the segment. For the other three calibration movements (Tilted to Stand, Extension to Stand, and Calf Raise to Squat), PCA was performed on the gyroscope data. Each of the calibration movements was intended to involve significant rotation about the sagittal axis of each segment with minimal rotation about the transverse and coronal axes. Therefore, the first principal component vector was assumed to be parallel to the sagittal axis of the segment. For all six functional calibration movements and the seated static calibration position, the coronal axis of each segment was estimated by taking the cross-product of the estimated transverse (from the standing static trial) and sagittal axes. The three estimated anatomical axes in the IMU reference frame were converted to a change-of-basis matrix that represented the orientation of the anatomical coordinate system in the reference frame of the IMU.

All gyroscope data from the functional tasks were filtered using a lowpass fourth-order Butterworth filter with a cutoff frequency of 5 Hz. The specific filter and cutoff frequency were selected based on previously reported filters used for IMU-based lower limb kinematics estimations [5]. The filtered data were translated from the sensor reference frame to the anatomical reference frame using the change-of-basis matrices determined by the calibration movements. The rotation of each segment during a functional task was then taken by integrating the transformed gyroscope data. Prior to data collection, the IMU gyroscopes were observed to include a constant non-zero bias, which produces a linear drift in the estimated sensor orientation when integrated over time. Therefore, gyroscopic drift was minimized by assuming the orientation of each segment was the same at the beginning and end of each task and then subtracting the linear drift from the calculated rotation angles to match this assumption. The resulting sagittal, transverse, and coronal rotation angles represented the rotation of a given segment relative to the initial position at the start of the task. By assuming all joint angles were zero at the beginning of each task due to the neutral standing posture, joint angles were then estimated by comparing the relative rotation of two segments adjacent to the joint by subtracting the rotation angles of the distal segment from the rotation angles of the proximal segment.

### 2.5. Data Analysis

The accuracy of each calibration method to measure kinematics was assessed by comparing the joint angle range of motion measured by the IMUs (IMU-ROM) to the range of motion measured by the motion capture system (MOCAP-ROM). The mean range of motion of pelvis orientation and joint angles of the hips, knees, and ankles were computed for each plane across all functional tasks. Following significant Shapiro–Wilk tests for normality, Wilcoxon signed-rank tests were performed to identify differences between IMU-ROM and MOCAP-ROM for all kinematic variables in all planes. The significance level (α) was set to 0.05 and was adjusted to 0.007 using a Bonferroni correction to account for comparing each of the seven calibration methods to motion capture independently. Differences exceeding 3° were considered clinically significant [24].

## 3. Results

Differences in range of motion are reported as the mean difference between IMU-ROM and MOCAP-ROM. A positive mean difference indicates that IMU-ROM was measured to be greater than MOCAP-ROM for a given kinematic variable. Mean differences in ROM for all kinematic variables for the Gait task are presented in Table 2, with coronal plane mean differences in ROM visualized in Figure 3. All other tasks are summarized in Appendix A.

### 3.1. Gait

Clinically significant differences between IMU-ROM and MOCAP-ROM were identified in the sagittal plane at the knee using the Slow Gait calibration method (mean difference: −3.67° ± 5.51 °) and at the hip using the Tilted to Stand calibration method (3.21° ± 3.25°). Although not clinically significant, other sagittal plane differences were identified at the pelvis for the Slow Gait calibration method (2.13° ± 2.95°), as well as at the pelvis (1.78° ± 2.61°) and hip (2.66° ± 2.89°) for the Fast Gait calibration method. The Gait, Seated Static, and Calf Raise to Squat calibration methods did not result in any differences in the sagittal plane. In the coronal plane, all calibration methods produced clinically significant differences at the knee and ankle, with the largest differences seen from the Calf Raise to Squat (16.58° ± 6.68°) and Slow Gait (24.11° ± 11.67°) calibration methods, respectively. Differences in the transverse plane at the hip and knee were observed for all calibration methods, and only the Fast Gait calibration method did not result in a difference at the ankle.

### 3.2. Overhead Squat

All calibration methods except Slow Gait resulted in significant differences in the sagittal plane at the hip (19.97° to 23.91°) and knee (11.47° to 14.92°). The three gait-based calibration methods produced differences at the pelvis (3.66° to 7.93°) and hip (11.16° to 19.73°) in the coronal plane, with Slow Gait and Gait also producing differences at the ankle (6.24° ± 6.40° and 5.11° ± 7.02°, respectively). No differences in the coronal plane were identified from the Seated Static, Tilted to Stand, Extension to Stand, or Calf Raise to Squat calibration methods; however, these four calibration methods displayed differences at the hip in the transverse plane (−6.18° to −11.39°).

### 3.3. Heel Touch

In the sagittal plane, differences were identified at the hip and knee for all calibration methods except Slow Gait, with the greatest differences produced by the Calf Raise to Squat (7.18° ± 4.88°) and Tilted to Stand (6.55° ± 3.63°) calibrations, respectively. The Slow Gait, Fast Gait, and Tilted to Stand calibrations also resulted in differences at the pelvis in the sagittal plane, although these differences were not clinically significant. No differences in the coronal plane were found from the Fast Gait, Tilted to Stand, or Extension to Stand calibration methods, while the other four calibrations produced differences at the knee (4.11° to 5.39°). Additionally, the Slow Gait calibration method resulted in a difference at the ankle (4.75° ± 5.58°). Differences in the transverse plane were found at the knee for all calibration methods (4.58° to 13.70°), with all but the Slow Gait calibration method showing differences at the hip (6.58° to 10.35°).

### 3.4. Single Leg Hop

The Fast Gait calibration method produced a significant difference in the sagittal plane at the knee (−5.37° ± 5.80°), and no other differences were identified in the sagittal plane across all calibration methods. In the coronal and transverse planes, each of the three gait-based calibration methods resulted in differences at the ankle (coronal: 6.04° to 11.02°; transverse: 5.88° to 9.02°) while every calibration method produced differences at the knee (coronal: 8.12° to 15.58°; transverse: 7.55° to 16.08°). Additionally, all calibration methods resulted in differences at the hip in the transverse plane (5.99° to 9.39°). No significant differences were detected at the pelvis in any plane across all calibration methods.

### 3.5. Drop Vertical Jump

Differences at the pelvis (6.41° to 7.69°), hip (20.90° to 23.57°), and knee (4.69° to 8.92°) in the sagittal plane were identified for the Fast Gait, Seated Static, Tilted to Stand, Extension to Stand, and Calf Raise to Squat calibration methods. All gait-based calibration methods produced differences at the ankle (−4.86° to −7.53°), with the Gait calibration also producing a difference at the hip (18.58° ± 14.37°). In the coronal plane, all calibration methods except the Calf Raise to Squat resulted in differences at the hip (4.73° to 20.99°) and ankle (16.51° to 21.97°). The Calf Raise to Squat calibration method resulted in one difference in the coronal plane at the knee (9.42° ± 11.35°). All calibration methods produced significant differences at the knee in the transverse plane (4.58° to 13.70°), with differences at the ankle additionally measured for the Gait (−4.13° ± 4.77°), Tilted to Stand (−4.86° ± 5.92°), and Extension to Stand (−4.54° ± 4.15°) calibrations.

### 3.6. Lateral Shuffle

No differences were measured in the sagittal plane from the Fast Gait or Extension to Stand calibration methods, with all other calibration methods producing differences at the ankle (−7.82° to −10.25°). In the coronal plane, differences were measured at the ankle using the Fast Gait (−5.98 ± 7.91°), Tilted to Stand (−5.14 ± 6.43), Extension to Stand (−5.74 ± 6.46), and Calf Raise to Squat (−4.27° ± 4.97°) calibration methods. All calibration methods resulted in differences at the knee in the coronal (7.79° to 10.20°) and transverse (5.79° to 11.20°) planes, with the greatest differences produced by the Calf Raise to Squat (10.20° ± 7.92°) and Fast Gait (11.20° ± 8.15°) calibrations, respectively.

### 3.7. Deceleration

All gait-based calibration methods produced differences at the knee in the sagittal plane (−6.08° to −10.24°). Differences were also seen in the sagittal plane at the ankle for all calibration methods (−8.26° to −12.86°). All calibration methods produced significant differences at the knee in the coronal plane (5.80° to 8.35°). In the transverse plane, all calibration methods resulted in differences at the knee (5.43° to 8.72°), and all gait-based calibrations produced differences at the hip as well (5.80° to 8.45°).

### 3.8. 45° Cut

The Slow Gait calibration method did not result in any differences in the sagittal plane, with all other calibrations producing differences at the hip (−7.23° to −10.57°). In the coronal plane, differences were calculated at the knee for all calibration methods (8.32° to 18.21°). All calibration methods except Extension to Stand resulted in differences at the knee in the transverse plane (6.89° to 12.27°). The Seated Static (9.87° ± 11.96°), Tilted to Stand (8.11° ± 8.27°), Extension to Stand (8.95° ± 9.78°), and Calf Raise to Squat (10.13° ± 9.41°) calibrations produced differences at the ankle in the transverse plane, with the Seated Static additionally producing a difference at the hip (−9.78° ± 10.05°).

## 4. Discussion

The purpose of this study was to investigate the accuracy of the lower limb range of motion calculated from wireless IMUs during sports-specific tasks using a variety of previously described functional and static sensor-to-segment calibration methods. The findings of this study show that the accuracy of each calibration method is dependent on the functional task as well as the specific joint angle being measured, and as a result, no single calibration method performs the best across all tasks.

Prior studies have investigated the accuracy of IMUs for lower limb joint kinematics during gait and other relatively simple movements [13,15,16,17,18,28]. Specifically, Lebleu et al. tested the same sensor-to-segment calibration methods used in this study during straight-line gait and several other more complex gait movements (e.g., stepping over an obstacle, ascending and descending stairs) with reported ROM discrepancies between 0.2° and 3.4° across all joints and planes of movement [13]. For the gait task in the current study, absolute ROM discrepancies varied between 0.3° and 14.5°, with the majority of values less than the threshold for clinical significance of 3.0°. The greater differences in ROM reported in the current study may be related to the methods used for computing joint angles following sensor-to-segment calibration. Simple integration of the gyroscope signal was used in this study while Lebleu et al. and other groups have utilized more complex methods such as attitude and heading reference systems (AHRS), which combine both the gyroscope and acceleration data from the IMUs. Compared to AHRS and other methods of sensor fusion, simple gyroscope integration has been reported to result in drift in the orientation estimation of an IMU over time [29]. Although several aspects of data collection and processing in this study were intended to reduce the effect of gyroscope drift, this is a likely source of additional error in the reported IMU-ROM values. Furthermore, Lebleu et al. and other groups have mounted the IMUs to rigid clusters of markers, rather than placing the sensors directly on the skin as was performed in the current study. While this methodology is likely to reduce differences in measured ROM between the IMUs and motion capture, it is not representative of practical applications of IMUs to measure kinematics outside of a motion capture lab setting. For example, many wireless IMUs such as the Delsys Trigno Avanti sensors used in this study are also capable of recording surface electromyography (EMG). The sensors would therefore require placement directly on muscles of interest if the user is interested in both kinematics and EMG metrics, both of which are commonly desired in sports injury prevention and recovery assessments.

To our knowledge, there is no prior work evaluating sensor-to-segment calibration methods for use in lower limb kinematic assessments during highly dynamic sports-specific tasks. Across all tasks, absolute differences between IMU-ROM and MOCAP-ROM varied between 0.1° and 20.8°. Apart from the Drop Vertical Jump task, mean differences between IMU-ROM and MOCAP-ROM were less than 3.0° for a majority of kinematic variables for all tasks. Additionally, of the 12 total variables measured (pelvis, hip, knee, and ankle ROM in all three planes), each task had between 9 and 12 variables for which at least one calibration method resulted in no significant differences in ROM. Large differences between IMU-ROM and MOCAP-ROM were identified in the sagittal plane for several tasks. This is likely a result of increased drift in the gyroscope signal that is caused by the relatively high motion involved with these tasks. Conversely, large ROM differences were also measured in the coronal and transverse planes for tasks with relatively little motion in these planes. These large differences may be due to a low signal-to-noise ratio as a result of the rapid motion inherent in sports-related movements. These results suggest that most lower limb range of motion variables can be adequately estimated using one or more sensor-to-segment calibration methods; however, the specific calibration method used may need to be selected based on the specific joints and planes of motion of interest for a given study.

The methods used for this study include several limitations of note. First, while the purpose of this study was to evaluate sensor-to-segment calibration methods, the results are also dependent on the method of joint angle estimation used following sensor calibration. The relatively simple gyroscope integration model used in this study was selected to minimize sources of error that are inherent in more complex models such as manually tuned filter parameters. Additionally, this model does not require the use of sensor hardware specifications, allowing it to be applied to a variety of IMU devices. Because the method of joint angle estimation used in this study is one of several previously described methods, it is likely that using an alternate method would influence the measured accuracy of each sensor-to-segment calibration method in measuring the lower-extremity range of motion. Future studies should investigate the effects of different joint angle estimation methods on the measured accuracy of IMU-ROM compared to motion capture during highly dynamic sports movements. Second, given that the magnetometer data from the IMUs were not accessible, it was not possible to establish the orientation of each sensor in a shared global reference frame. As a result, it was assumed that all sensor reference frames were aligned at the start and end of each trial. This required each participant to assume an identical neutral standing position before and after the completion of each task. Any variations in segment orientation between the two standing positions introduced errors to the gyroscope integration and drift mitigation methods used in this study. Although each participant understood this requirement and performed each task as instructed, it is very likely that this assumption resulted in decreased accuracy of the measured IMU-ROM. Additionally, the application of this assumption as well as the functional sensor-to-segment calibration methods in a clinical setting may prove challenging for patients with limited mobility.

## 5. Conclusions

Wearable sensors present methods of measuring lower-limb kinematics in an ambulatory setting when motion capture is not possible. This study demonstrates that previously established sensor-to-segment functional and static calibration methods can be applied to sports-specific kinematic assessments to measure lower limb joint angle range of motion in all three planes of motion. Each calibration method demonstrated varying levels of accuracy for each task and kinematic variable, and thus, the specific method used for a given study should be selected depending on the specific movement and joint angles of interest. Further work is required to fully establish the most accurate sensor calibration and joint angle estimation methods for sports-specific kinematic assessments, and the results presented here are intended to provide some guidance for future investigations.

## Figures and Tables

**Figure 1 sensors-23-07987-f001:**
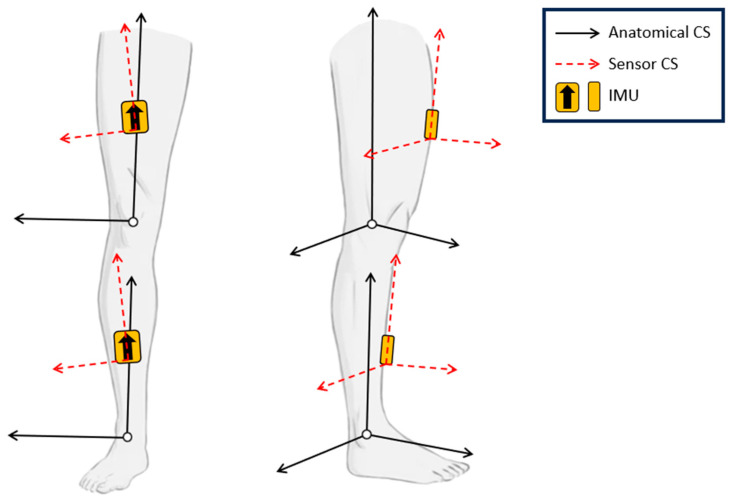
Coordinate systems (CS) for the anatomical (black) and IMU (red dashed) reference frames of the thigh and shank segments. The sensor coordinate system may not be aligned with the anatomical coordinate system due to variability in sensor placement.

**Figure 2 sensors-23-07987-f002:**
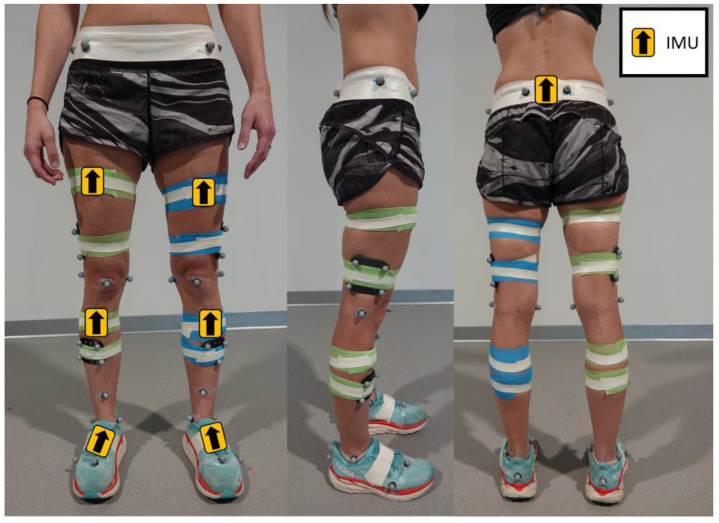
Placement of seven Delsys Trigno Avanti IMU sensors: Sacrum, anterior thighs, anterior shanks, and the dorsal side of the feet. All sensors were secured with athletic tape to minimize motion artifacts during functional tasks.

**Figure 3 sensors-23-07987-f003:**
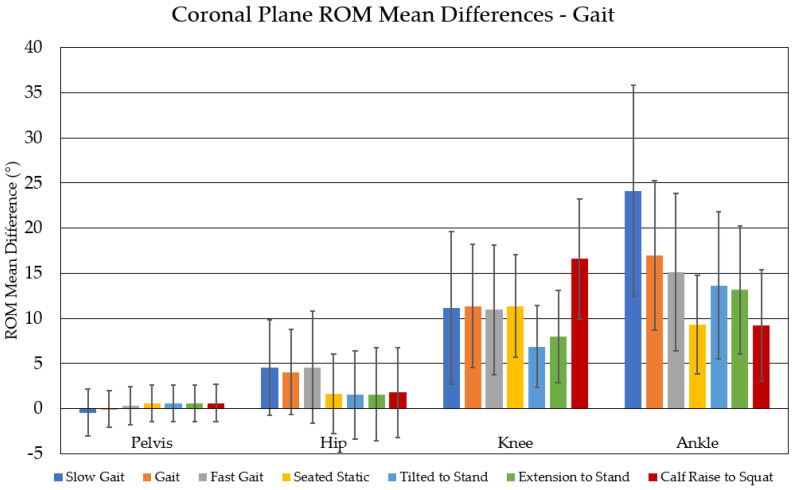
Mean differences (±SD) in ROM between motion capture and IMUs using each calibration method for the gait task. Positive mean differences indicate higher ROM measured by the IMUs relative to motion capture.

**Table 1 sensors-23-07987-t001:** Sensor-to-Segment Calibration Movement Instructions.

Calibration Movement	Verbal Instruction
Standing Static	Assume a neutral standing position with your feet flat on the floor and toes pointing forward.
Seated Static	Sit on the chair in a leaned-back position with your legs extended straight in front of you and toes pointing up.
Slow Gait	Walk slowly to the other end of the lab.
Normal Gait	Walk at a normal pace to the other end of the lab.
Fast Gait	Walk quickly to the other end of the lab as if you were late to a meeting.
Tilted to Stand	Start seated in a leaned-back position with extended legs and toes pointing up. Bend your knees, lean forward, stand up and stop moving.
Extension to Stand	Start in a neutral seated position with knees bent and feet flat on the floor. Extend your legs out in front of you, bring them back to the position you started in, lean forward, stand up and stop moving.
Calf Raise to Squat	Start in a neutral standing position. Raise up on your toes, return to neutral standing, squat down, and return to neutral standing.

**Table 2 sensors-23-07987-t002:** Mean differences (±SD) between IMU-ROM and MOCAP-ROM for the Gait task for all calibration methods.

Plane	Joint	Slow Gait	Gait	Fast Gait	Seated Static	Tilted to Stand	Extension to Stand	Calf Raise to Squat
Sagittal	Pelvis	**2.13 ± 2.95**	1.65 ± 3.05	**1.78 ± 2.61**	0.67 ± 2.35	0.79 ± 2.36	0.71 ± 2.45	1.02 ± 2.33
	Hip	1.21 ± 3.36	1.66 ± 2.98	**2.66 ± 2.89**	1.84 ± 3.13	**3.21 ± 3.25**	**2.89 ± 3.64**	2.46 ± 3.95
	Knee	**−3.67 ± 5.51**	−0.81 ± 3.63	−1.64 ± 3.54	0.14 ± 2.70	−0.55 ± 2.42	0.13 ± 2.97	−2.78 ± 6.71
	Ankle	0.17 ± 8.83	0.38 ± 5.36	−0.62 ± 5.27	0.25 ± 4.40	0.68 ± 4.54	0.70 ± 4.54	2.48 ± 6.35
Coronal	Pelvis	−0.46 ± 2.58	−0.01 ± 2.04	0.29 ± 2.10	0.58 ± 2.06	0.61 ± 2.03	0.61 ± 2.03	0.59 ± 2.08
	Hip	**4.57 ± 5.29**	**4.05 ± 4.75**	**4.58 ± 6.20**	1.66 ± 4.41	1.51 ± 4.92	1.57 ± 5.15	1.80 ± 4.98
	Knee	**11.16 ± 8.49**	**11.36 ± 6.80**	**10.94 ± 7.20**	**11.35 ± 5.67**	**6.87 ± 4.57**	**7.99 ± 5.10**	**16.58 ± 6.68**
	Ankle	**24.11 ± 11.67**	**16.97 ± 8.30**	**15.10 ± 8.74**	**9.26 ± 5.46**	**13.63 ± 8.15**	**13.14 ± 7.06**	**9.21 ± 6.13**
Transverse	Pelvis	−0.72 ± 2.56	−0.81 ± 1.89	−0.87 ± 2.03	−0.75 ± 1.92	−0.76 ± 1.95	−0.75 ± 1.91	−0.84 ± 2.09
	Hip	**7.55 ± 5.54**	**5.81 ± 5.22**	**5.00 ± 5.16**	**8.34 ± 5.40**	**5.78 ± 5.74**	**6.24 ± 5.97**	**6.45 ± 5.95**
	Knee	**9.11 ± 8.21**	**8.51 ± 8.78**	**10.83 ± 9.68**	**11.26 ± 6.14**	**8.01 ± 8.44**	**10.56 ± 8.91**	**15.33 ± 11.66**
	Ankle	**11.56 ± 10.50**	**6.71 ± 7.81**	3.92 ± 7.03	**6.25 ± 6.34**	**4.84 ± 5.45**	**5.21 ± 7.06**	**7.05 ± 9.20**

Note: Statistically significant differences are presented in bold.

## Data Availability

The data presented in this study are available upon request from the corresponding author.

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
