# Peer review of "Inertial Measurement Unit Sensor-to-Segment Calibration Comparison for Sport-Specific Motion Analysis"

_sensors, 2023, doi:10.3390/s23187987_

Round 1
Reviewer 1 Report
MDPI Sensors Reviewer Comments:
This study aims to evaluate sensor-to-segment calibration methods to measure joint angle range of motion. There are the following concerns, which author need to clarify and improve:
1. The introduction and state of art must be improved. Authors need to mention more related research and more references.
2. The data processing part is important part. Principal component analysis (PCA) was performed on the IMU data. However, authors did not describe in detail the working principle of PCA on IMU data and why PCA was selected in this case. Authors must expand and dive deeper into this part.
3. The relationship between human body frame and IMU sensor frame should be demonstrated more clearly in a specific picture.
4. The calibration technique should be depicted and shown in diagram or flow chart to be clearer.
5. The optical capture was used as gold standard comparison, so it is necessary to describe its working principle also.
6. Why the IMU technique can replace the optical technique?
7. Authors must describe how the angles were calculated from IMU data.
8. The results are shown only on the statistics. The charts need to be utilized for better visualization.
Authors must double-check the English grammar and format of the manuscript carefully.
Round 2
Reviewer 1 Report
Authors improved the article.
I have one last question for the author: the gyroscope includes noise and drift, which negatively impact the angle integration. How did author solve this issue?
Author should check the text, grammar and whole structure carefully before publishing the article.
Authors improved the article.
I have one last question for the author: the gyroscope includes noise and drift, which negatively impact the angle integration. How did author solve this issue?
Author should check the text, grammar and whole structure carefully before publishing the article.
